# Effects of COVID-19 disease on PAI-1 antigen and haematological parameters during disease management: A prospective cross-sectional study in a regional Hospital in Ghana

**Charles Nkansah**[1,2]*, **Michael Owusu**[3,4], **Samuel Kwasi Appiah**[1,2], **Kofi Mensah**[1,2], **Simon Bannison Bani**[5], **Felix Osei-Boakye**[6], **Lawrence Duah Agyemang**[3,7], **Ezekiel Bonwin Ackah**[3], **Gabriel Abbam**[1], **Samira Daud**[1], **Yeduah Quansah**[5], **Charles Angnataa Derigubah**[8], **Francis Atoroba Apodola**[9], **Valentine Ayangba**[10], **David Amoah Afrifa**[3,11], **Caleb Paul Eshun**[5], **Abdul-Waliu Iddrisu**[5], **Selina Mintaah**[3,12], **Benjamin Twum**[13], **Abidatu Mohammed**[5], **Eugene Mensah Agyare**[5], **Wendy Akomeah Gyasi**[7,14], **Peace Esenam Agbadza**[5], **Candy Adwoa Ewusiwaa Wilson**[5], **Seth Anane**[7], **Prince Antwi**[15], **Reginald Akwasi Yeboah Antwi**[16]

**1** Department of Haematology, School of Allied Health Sciences, University for Development Studies, Tamale, Ghana, **2** Department of Medical Diagnostics, Faculty of Allied Health Sciences, Kwame Nkrumah University of Science and Technology, Kumasi, Ghana, **3** Department of Medical Laboratory Sciences, Faculty of Health Science and Technology, Ebonyi State University, Abakaliki, Nigeria, **4** Kumasi Centre for Collaborative Research, Kumasi, Ghana, **5** Department of Biomedical Laboratory Sciences, School of Allied Health Sciences, University for Development Studies, Tamale, Ghana, **6** Department of Medical Laboratory Technology, Faculty of Applied Science and Technology, Sunyani Technical University, Sunyani, Ghana, **7** Department of Clinical Microbiology, Laboratory Service Directorate, Komfo Anokye Teaching Hospital, Kumasi, Ghana, **8** Department of Medical Laboratory Technology, School of Applied Science and Arts, Bolgatanga Technical University, Bolgatanga, Ghana, **9** Department of Medical Diagnostics, Faculty of Allied Health Sciences, College of Nursing and Allied Health Sciences, Nalerigu, Ghana, **10** Department of Nursing and Midwifery, Faculty of Nursing and Midwifery, College of Nursing and Allied Health Sciences, Nalerigu, Ghana, **11** Department of Medical Laboratory, Ankaase Methodist Hospital, Kumasi, Ghana, **12** Department of Haematology, Laboratory Service Directorate, Komfo Anokye Teaching Hospital, Kumasi, Ghana, **13** Department of Medical Laboratory, Sunyani Regional Hospital, Sunyani, Ghana, **14** Department of Theoretical and Applied Biology, Faculty of Biosciences, Kwame Nkrumah University of Science and Technology, Kumasi, Ghana, **15** Department of Microbiology, School of Health and Life Sciences, TEESSIDE University, Middlesbrough, United Kingdom, **16** Medical Laboratory Department, Asafo Boakye Specialist Hospital, Kumasi, Ghana

* cnkansah86@yahoo.com

## Abstract

### Background

Individuals with COVID-19 experience thrombotic events probably due to the associated hypofibrinolysis resulting from the upregulation of plasminogen activator inhibitor-1 (PAI-1) antigen. This study evaluated plasma PAI-1 antigen levels and haematological parameters before treatment and after recovery from severe COVID-19 in Ghana.

### Materials and methods

This cross-sectional study was conducted at Sunyani Regional Hospital, and recruited 51 patients who had RT-PCR-confirmed SARS-CoV-2. Participants' sociodemographic data

**Data Availability Statement:** Data underlying our findings is deposited at Flow Repository (ID: FR-FCM-Z6Z7).

**Funding:** The authors received no specific funding for this work.

**Competing interests:** The authors have declared that no competing interests exist.

and clinical characteristics were taken from the hospital records. Venous blood was taken before COVID-19 treatment commenced for FBC, PAI-1 and ferritin assays. FBC was assessed using an automated haematology analyzer, whilst plasma PAI-1 Ag and serum ferritin levels were assessed with sandwich ELISA. All the tests were repeated immediately after participants recovered from COVID-19.

## Results

Of the 51 participants recruited into the study, 78.4% (40) had non-severe COVID-19 whiles 21.6% (11) experienced a severe form of the disease. Severe COVID-19 participants had significantly lower haemoglobin (g/dL): 8.1 (7.3–8.4) vs 11.8 (11.0–12.5), $p<0.001$; RBC x $10^{12}$/L: 2.9 (2.6–3.1) vs 3.4 (3.1–4.3), $p = 0.001$; HCT%: 24.8 ± 2.6 vs 35.3 ± 6.7, $p<0.001$ and platelet x $10^9$/L: 86.4 (62.2–91.8) vs 165.5 (115.1–210.3), $p<0.001$, compared with the non-severe COVID-19 group. But WBC x $10^9$/L: 11.6 (9.9–14.2) vs 5.4 (3.7–6.6), $p<0.001$ and ferritin (ng/mL): 473.1 (428.3–496.0) vs 336.2 (249.9–386.5), $p<0.001$, were relatively higher in the participants with severe COVID-19 than the non-severe COVID-19 counter-parts. Also, the severely ill SARS-CoV-2-infected participants had relatively higher plasma PAI-1 Ag levels (ng/mL): 131.1 (128.7–131.9) vs 101.3 (92.0–116.8), $p<0.001$, than those with the non-severe form of the disease. Participants had lower haemoglobin (g/dL): 11.4 (8.8–12.3 vs 12.4 (11.5–13.6), $p<0.001$; RBC x $10^{12}$/L: 3.3 (2.9–4.0) vs 4.3 (3.4–4.6), $p = 0.001$; absolute granulocyte count x $10^9$/L: 2.3 ± 1.0 vs 4.6 ± 1.8, $p<0.001$, and platelet x $10^9$/L: 135.0 (107.0–193.0) vs 229.0 (166.0–270.0), $p<0.001$ values at admission before treatment commenced, compared to when they recovered from the disease. Additionally, the median PAI-1 Ag (ng/mL): 89.6 (74.9–100.8) vs 103.1 (93.2–128.7), $p<0.001$ and ferritin (ng/mL): 242.2 (197.1–302.1) vs 362.3 (273.1–399.9), $p<0.001$ levels were reduced after a successful recovery from COVID-19 compared to the values at admission.

## Conclusion

Plasma PAI-1 Ag level was higher among severe COVID-19 participants. The COVID-19-associated inflammation could affect red blood cell parameters and platelets. Successful recovery from COVID-19, with reduced inflammatory response as observed in the decline of serum ferritin levels restores the haematological parameters. Plasma levels of PAI-1 should be assessed during the management of severe COVID-19 in Ghana. This will enhance the early detection of probable thrombotic events and prompts Physicians to provide interventions to prevent thrombotic complications associated with COVID-19.

## Introduction

The world has experienced the debilitating effects of Coronavirus disease-2019 (COVID-19) since it emerged from Wuhan, China, claiming millions of lives. Globally, as of 14 November 2022, 635,443,811 total confirmed cases and 6,604,811 deaths had been recorded. Around the same period, Ghana reported 170,996 overall confirmed cases and 1,461 deaths [1]. The causative agent for COVID-19 is Severe Acute Respiratory Syndrome Coronavirus-2 (SARS-CoV-2), and this virus interacts with angiotensin-converting enzyme-2 (ACE-2) in the lungs and other parts of the body to cause pulmonary and systemic complications [2, 3]. Fever, dyspnea,

anosmia, itchy throat, and dry cough are common clinical symptoms associated with COVID-19 progression [4, 5].

The deadly pandemic has been reported to trigger a systemic cytokine storm, due to its tendency to induce inflammation leading to haematological complications, endothelial dysfunction, organ failure and death [6, 7]. Abdullah and colleagues [6] reported the occurrence of anaemia, neutrophilia, lymphopenia, monocytosis, eosinopenia, moderate thrombocytopenia, and less commonly thrombocytosis among individuals with COVID-19. The associated inflammation and subsequent rapid increased release of excessive cytokines (cytokine storm) especially tumour necrosis factor-α (TNF-α), interleukin (IL)-1 and 6, contribute to coagulopathies, with enhanced activation of procoagulants and likely hypofibrinolysis. The disturbed fibrinolysis among SARS-CoV-2 infected persons may result from the upregulation of PAI-1, the main inhibitor of the fibrinolytic pathway [7].

PAI-1 is a serine protease inhibitor with molecular weight ranging from 38 to 70 kDa (depending on the level of glycosylation). The enzyme is synthesized by endothelial cells, hepatocytes, vascular smooth muscle cells, mesangial cells, fibroblasts, monocytes/macrophages, and adipose tissue-derived stromal cells [8, 9]. PAI-1 is thought to be an independent marker of thrombosis, and the protein is upregulated in inflammatory processes [10].

Thrombotic complications especially severe pulmonary embolisms were identified among COVID-19 patients in the Intensive Care Units (ICUs), even though patients received high-dose prophylaxis [11, 12]. About 27% of severely ill COVID-19 patients admitted to the ICU developed venous thromboembolism while 4.45% experienced arterial thrombotic complications [13]. Another study also reported a highly variable prevalence of deep vein thrombosis (DVT) between 2.0% and 14.8% in ICU patients [3]. The Nougier and co. study reported the occurrence of unexpected deaths, high morbidity, and significantly high D-dimer levels in COVID-19 patients during the disease's advancement [14].

The associated inflammation during COVID-19 has been identified by previous studies where they confirmed elevated levels of ferritin (an acute phase protein), usually coupled with the presence of anaemia [15–19].

The few studies that have evaluated PAI-1 among COVID-19 patients made a one-time assessment and did not consider the changes that may occur during the recovery period. Also, regardless of the significant role PAI-1 could play in the development of thrombotic complications during COVID-19 progression, the PAI-1 enzyme has not been studied in Ghana, based on our knowledge. Early detection and control of thrombotic-related factors could be beneficial to avert further complications of the COVID-19 disease in the infected individuals. This study evaluated plasma PAI-1 Ag levels before treatment and after recovery from severe and non-severe COVID-19 in the Middle Belt of Ghana.

## Materials and methods

### Study design and setting

This was a prospective cross-sectional study that was conducted from January to May, 2022 at the Sunyani Regional Hospital, Bono Region, Ghana (Fig 1). The hospital serves as a referral facility for most health centres in the Bono and Ahafo Regions. Sunyani is the capital city of the Bono Region and is located in west-central Ghana with geographical coordinates 7°20′N 2°20′W. The forest—surrounding city has over 200,000 inhabitants with 48% engaged in agricultural-related activities [20, 21]. The state-of-the-art regional hospital was established over nine decades ago and provides general and specialized medical services. The facility has a well-functioning medical laboratory unit and was designated as one of the test centres for SARS-CoV-2 in Ghana [22].

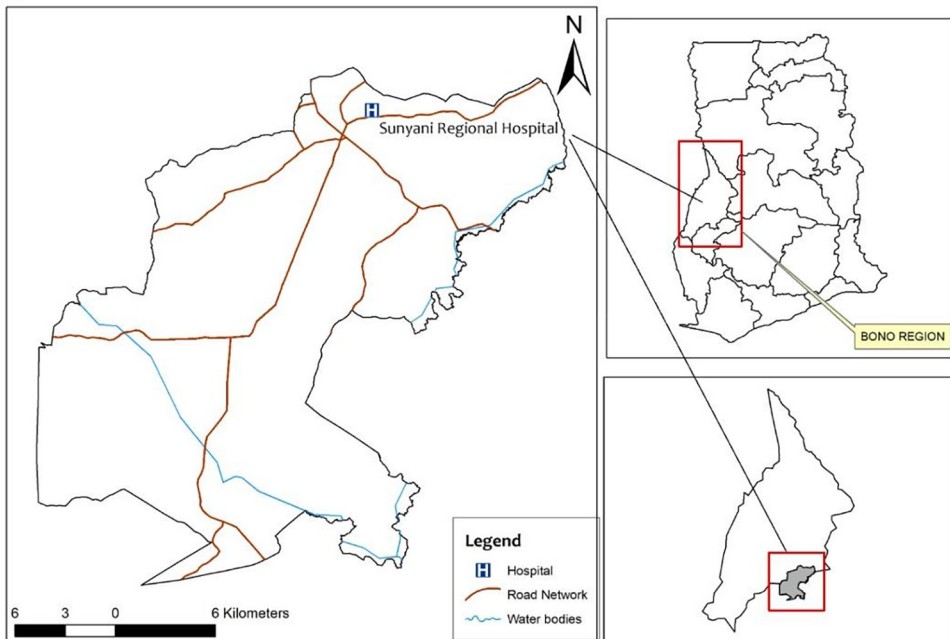

**Fig 1. A modified map of Ghana created with ArcGIS, depicting Sunyani Regional Hospital (study site) in the Bono Region.** Source of the basemap: https://data.gov.gh/dataset/shapefiles-all-districts-ghana-2012-216—260districts.

## Ethical considerations

The study adhered to the principles stated in the Helsinki Declaration on research involving humans. The Committee on Human Research, Publication and Ethics of Kwame Nkrumah University of Science and Technology (KNUST), Kumasi, (CHRPE/AP/012/22) approved the study. Permission was obtained from the management of the Sunyani Regional Hospital. Informed consent was obtained from the study participants after signing or thumb-printing the consent form before their inclusion into the study.

## Study population

Reverse transcriptase polymerase chain reaction (RT-PCR) confirmed SARS-CoV-2 positive patients, both severe and non-severe receiving management at the Sunyani Regional Hospital during the study period were selected for the study.

## Sample size determination

The Cochran formula was used to determine the sample size.

$$n = \frac{Z^2 P(1-P)}{d^2}$$

Where:

$z$ = 1.96 at 95% confidence interval (CI).

$p$ = prevalence of deep vein thrombosis in COVID-19, being 3% [23].

$d$ = margin of error, at 95% CI, value is 0.05.

$n$ = minimum sample size required.

$$n = \frac{(1.96)^2 \times 0.03(1-0.03)}{(0.05)^2}$$

$$n = (3.8416 \times 0.0291)/(0.0025)$$
$$n = 44.72$$
$$n \approx 45$$

The minimum number of participants required for this study was 45. However, this study recruited 51 RT-PCR-confirmed COVID-19 subjects (40 non-severe and 11 severe COVID-19 patients).

### Study participants selection criteria

In this study, RT-PCR-confirmed COVID-19 patients who visited the hospital during the study period were selected as participants.

### Inclusion criteria

All individuals who tested RT-PCR positive for COVID-19 and gave their informed consent were included in the study.

### Exclusion criteria

COVID-19 patients with known chronic diseases, those on medications or with other haematological abnormalities were excluded from this study. Also, participants who refused to give their informed consent were excluded from the study.

### Data collection and sources of data

Participants' socio-demographic and clinical characteristics were obtained from the hospital records. Also, nasopharyngeal swabs and blood samples were taken from the participants using standard aseptic techniques for laboratory assay.

### Sample collection and processing

Nasopharyngeal samples were collected into sterile tubes containing a virus transport medium, for the detection of SARS-CoV-2. Immediately a COVID-19 patient was diagnosed by RT-PCR, after the consent, seven (7) mL of whole blood was collected aseptically and dispensed into various tubes. Two (2) mL into purple top vacutainers (containing $K_2EDTA$ anticoagulant) for full blood counts (FBC), 3 mL into sodium citrate tubes for analysis of plasma PAI-1 antigen and 2 mL into a plain tube for ferritin assessment. The blood was adequately mixed with the $K_2EDTA$ and sodium citrate anticoagulants in the respective tubes to prevent clotting. The tubes were then uniquely code-labelled to ensure the anonymity of participants. Blood samples in the plain tube (after sufficient clotting) and citrate tube were centrifuged (CENTRIFUGE 80–1, Japan) at 3000 rpm for 15 minutes to separate the respective serum and plasma from the cells. The plasma and serum components were aliquoted into Eppendorf tubes and stored at -20°C for plasma PAI-1 antigen and serum ferritin levels batch analysis using the ELISA technique.

Once a participant gained full recovery from the COVID-19, after showing no clinical sign and no RT-PCR positive result, the steps above were repeated for blood sample collection, processing, storage and batch analyses for plasma PAI-1 Ag and serum ferritin. FBC test was performed on daily basis following sample collection into $K_2EDTA$ tubes.

### Detection of SARS-CoV-2

The SARS-CoV-2 genome was detected from the nasopharyngeal swab samples, using RT-PCR as recommended by the US Centers for Disease Control and Prevention (CDC) [24,

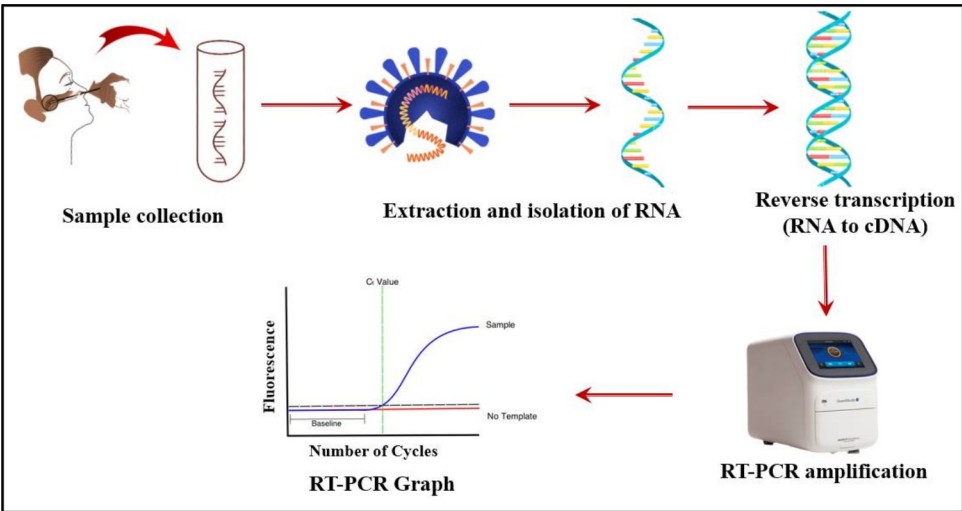

**Fig 2. Steps involved in RT-PCR for SARS-CoV-2 detection** [24, 25].

25]. The technique detects and amplifies two nucleoside coding genomic regions, N1 and N2, respectively. The procedures involved in the detection of the SARS-CoV-2 genome have been summarized below (Fig 2). The test was carried out at the COVID Laboratory, Sunyani Regional Hospital. The severity of COVID-19 was classified based on the protocols recommended by the World Health Organization (WHO) [26]. Participants were diagnosed with non-severe COVID-19 when the nasopharyngeal swab contained SARS-CoV-2, but without any sign of severe or critical disease. Severe COVID-19 patients had SARS-CoV-2 and either oxygen saturation (SpO2) less than 90% on room air, signs of pneumonia or signs of severe respiratory distress. Patients tend to experience critical COVID-19 when in addition to the presence of SARS-CoV-2, there is either an urgent need for life-sustaining treatment, acute respiratory distress syndrome (ARDS), sepsis or septic shock [26].

## Assessment of full blood counts (FBC)

Whole blood samples were transported to the Haematology Laboratory of the Sunyani Regional Hospital, where FBC was performed on each day of sample collection. A fully automated three-part haematology analyser (Mindray BC3000 Plus, China) was used for the estimation of the FBC parameters. The Haematology analyser used two independent methods for the measurement of blood cells, namely, the Coulter and colorimetric methods [27].

The Coulter method employed electrical impedance and flow cytometry principles for its analysis. The blood cells were counted and differentiated by measuring the change in resistance produced by a cell suspended in a conductive diluent as it passed through an aperture of known dimensions. This change in resistance was recorded as frequency and amplitude, which respectively translated into the number and size/volume of the cell. In one channel/bath, red cells and platelets were counted, and in another channel white cells and their differentials are counted, sizing them into three sub-populations—lymphocytes (small-sized), mid-sized (mid or mixed) cells (comprising monocytes, basophils and eosinophils) and granulocytes (large-sized cells mainly neutrophils). The differentiation of red cells from platelets also arose from the different sizes of the two types of cells.

In the colorimetric method, the red cells were bubble mixed with cyanide-free sodium lauryl sulphate (SLS) reagent that lysed the cells and converted the haemoglobin molecules to

haemoglobin complexes that were measurable at a wavelength of 540 nm. The amount of hae-moglobin complexes was directly proportional to the number of haemoglobin molecules before the complex formation. All other parameters of FBC related to red cells and platelets were derived from a number of arithmetic combinations of the results from these blood cell measurements [27].

### Plasma plasminogen activator inhibitor-1 and serum ferritin assay using ELISA

Plasma PAI-1 Ag and serum ferritin levels were assayed by the sandwich ELISA method using commercially prepared ELISA kits (Biobase, China). All procedures were carried out by strictly following the manufacturer's instructions for each analyte. The ELISA plates were washed manually while the plasma concentrations of the analytes were determined using a microplate ELISA reader (Mindray MR-96A, China) at a wavelength of 450 nm. The ELISA laboratory investigations were done at Ankaase Methodist Hospital Laboratory, Ashanti Region, Ghana.

### Data analysis

Data generated were analyzed using IBM Statistical Package for the Social Sciences (SPSS) software version 26.0 (IBM Corp., Armonk, NY, USA). Normality was tested with one-sample Kolmogorov-Smirnov test and Shapiro-Wilk tests. Sample characteristics were summarized with descriptive statistics, and the data were presented in tables and figures. Parametric data (pulse, $SPO_2$%, blood pressure, HCT%, MCV, granulocytes and MPV) were presented in means ± standard deviation and non-parametric data (age, temperature, Hb, RBC, MCH, MCHC, RDW-CV, WBC, lymphocyte count, platelet, PDW%, PCT%, ferritin and PAI-1) in medians (1st-3rd quarters). Unpaired parametric data were compared with Student's T-test and unpaired non-parametric data were compared with Mann-Whitney U-test. Paired parametric data (on admission and after recovery) were compared using the Paired Sample T-test, whilst paired non-parametric data were compared with Wilcoxon signed-rank test. A $p < 0.05$ was considered statistically significant.

### Results

### Socio-demographic and clinical characteristics of the study participants stratified by COVID-19 severity

Of the 51 participants recruited into the study, the majority, 78.4% (40) had non-severe COVID-19 whiles 21.6% (11) experienced a severe form of the disease. The median age of the study participants was 34 (29–44) years, with 21 years and 81 years being the minimum and maximum years respectively. The majority of the participants were females, 66.7% (34), between the ages of 30–39 years, 45.1% (23), and only 17.6% (9) of them were 50 years and above. Temperature (°C): 38.7 (38.3–38.8) vs 37.5 (36.9–38.1), $p < 0.001$; pulse (bpm): 96.5 ± 11.4 vs 88.0 ± 7.3, $p = 0.004$; and SBP (mmHg): 153.0 ± 10.8 vs 124.6 ± 16.9, $p < 0.001$ were significantly higher in the severe COVID-19 patients compared with their counterparts with the non-severe form of the disease. However, SpO2 (%): 87.9 ± 3.4 vs 94.8 ± 3.1, $p < 0.001$, was lower in the severe COVID-19 participants than in the non-severe group, as shown in Table 1.

### Full blood counts and serum ferritin levels of the study participants stratified by COVID-19 severity

Haemoglobin (Hb) (g/dL): 8.1 (7.3–8.4) vs 11.8 (11.0–12.5), *$p < 0.001$*; red cell count (RBC) x $10^{12}$/L: 2.9 (2.6–3.1) vs 3.4 (3.1–4.3), *$p = 0.001$*; haematocrit (HCT)%: 24.8 ± 2.6 vs 35.3 ± 6.7,

**Table 1. Socio-demographic and clinical characteristics of the study participants stratified by COVID-19 severity.**

| Variables | Total Participants (N = 51) | COVID-19 Severity | | P-value |
|---|---|---|---|---|
| | | Non-severe (N = 40, 78.4%) | Severe (N = 11, 21.6%) | |
| Age (years) | 34 (29–44) | 34 (29.0–45.8) | 34 (29.0–38.0) | 0.882 |
| Age Category | | | | 0.391 |
| 20–29 | 27.5 (14) | 27.5 (11) | 27.3 (3) | |
| 30–39 | 45.1 (23) | 42.5 (17) | 54.5 (6) | |
| 40–49 | 9.8 (5) | 10.0 (4) | 9.1 (1) | |
| 50–59 | 5.9 (3) | 5.0 (2) | 9.1 (1) | |
| >59 | 11.7 (6) | 15.0 (6) | - | |
| Gender | | | | 1.000 |
| Male | 33.3 (17) | 32.5 (13) | 36.4 (4) | |
| Female | 66.7 (34) | 67.5 (27) | 63.6 (7) | |
| Vital Signs | | | | |
| Temperature (˚C) | 37.8 (37.0–38.3) | 37.5 (36.9–38.1) | 38.7 (38.3–38.8) | **<0.001** |
| Pulse (bpm) | 89.9 ± 8.9 | 88.0 ± 7.3 | 96.5 ± 11.4 | **0.004** |
| SpO2 (%) | 93.3 ± 4.2 | 94.8 ± 3.1 | 87.9 ± 3.4 | **<0.001** |
| Systolic BP (mmHg) | 133.1 ± 18.9 | 124.6 ± 16.9 | 153.0 ± 10.8 | **<0.001** |
| Diastolic BP (mmHg) | 96.0 ± 9.7 | 78.5 ± 9.7 | 84.5 ± 8.4 | 0.066 |

N = Number of participants; COVID-19 = Coronavirus disease-2019; ˚C = Degree Celsius; temperature, 36.1–37.2˚C; SpO2 = Oxygen saturation, 95–100%; SBP = Systolic Blood Pressure, 110–130 mmHg; Diastolic BP = Diastolic Blood Pressure, <80 mmHg; pulse, 60–100 bpm; bpm = Beats per minute; mmHg = Millimetres of Mercury. Categorical data were presented in frequencies with corresponding percentages in parentheses, and were appropriately compared with Fisher's exact test and one-way ANOVA. Student's T-test was used to compare parametric data (presented in mean ± standard deviation), and non-parametric data (presented in median (25$^{th}$-75$^{th}$ percentiles)) were compared with Mann Whitney U-test. p<0.05 was considered statistically significant.

$p<0.001$; mean cell volume (MCV) (fL): 88.9 ± 11.4 vs 97.5 ± 11.4, $p = 0.041$; and platelet count (PLT) x $10^9$/L: 86.4 (62.2–91.8) vs 165.5 (115.1–210.3), $p<0.001$, were significantly lower in the severe COVID-19 participants compared to those with non-severe COVID-19. However, severe COVID-19 participants had significantly higher levels of ferritin ((ng/mL)) than their counterparts with the non-severe form of the disease: 473.1 (428.3–496.0) vs 336.2 (249.9–386.5), $p<0.001$ (Table 2).

## Plasma plasminogen activator inhibitor-1 antigen levels and COVID-19 severity among the study participants

Fig 3 shows the plasma plasminogen activator inhibitor-1 (PAI-1) antigen (Ag) levels stratified by COVID-19 severity of the study participants. The median PAI-1 Ag level among the fifty-one COVID-19 participants in this study was 103.1 ng/mL (93.2–128.7). The median PAI-1 Ag level was relatively higher in the participants with severe COVID-19 than those with the non-severe form of the disease: 131.1 (128.7–131.9) vs 101.3 (92.0–116.8), $p<0.001$ (Fig 3).

## Changes in full blood counts and ferritin levels before treatment and at recovery

Table 3 shows the changes in FBC parameters and ferritin levels during COVID-19 management at Sunyani Regional Hospital. There were significant changes in FBC parameters before treatment commenced and at recovery from COVID-19. Respectively, Hb, RBC, HCT, Gran#, PLT, PDW and PCT were significantly increased after successful recovery compared to the

**Table 2. Full blood counts and serum ferritin levels of the study participants stratified by COVID-19 severity.**

| Variables | Total Participants (N = 51) | COVID-19 Severity | | P-value |
|---|---|---|---|---|
| | | Non-severe (N = 40) | Severe (N = 11) | |
| Hb (g/dL) | 11.4 (8.8–12.3) | 11.8 (11.0–12.5) | 8.1 (7.3–8.4) | **<0.001** |
| RBC x $10^{12}$/L | 3.3 (2.9–4.0) | 3.4 (3.1–4.3) | 2.9 (2.6–3.1) | **0.001** |
| HCT% | 33.0 ± 7.4 | 35.3 ± 6.7 | 24.8 ± 2.6 | **<0.001** |
| MCV (fL) | 97.5 ± 11.4 | 97.5 ± 11.4 | 88.9 ± 11.4 | **0.041** |
| MCH (pg) | 32.5 (28.3–37.1) | 34.2 (29.0–37.7) | 32.4 (27.1–34.4) | 0.053 |
| MCHC (g/dL) | 34.5 (33.7–35.8) | 34.6 (33.6–35.8) | 34.4 (34.1–35.5) | 0.936 |
| RDW-CV% | 14.6 (13.6–15.6) | 14.3 (13.3–15.2) | 15.4 (14.6–19.9) | **0.023** |
| TWBC x $10^9$/L | 5.7 (4.2–9.7) | 5.4 (3.7–6.6) | 11.6 (9.9–14.2) | **<0.001** |
| Lymph# x $10^9$/L | 3.3 (2.1–6.2) | 2.8 (1.9–3.7) | 7.6 (6.5–10.3) | **<0.001** |
| Gran# x $10^9$/L | 2.3 ± 1.0) | 2.0 ± 0.8 | 3.5 ± 0.8 | **0.001** |
| MID# x $10^9$/L | 0.4 (0.3–0.6) | 0.4 (0.3–0.6) | 0.6 (0.3–0.8) | 0.054 |
| PLT x $10^9$/L | 135.0 (107.0–193.0) | 165.5 (115.1–210.3) | 86.4 (62.2–91.8) | **<0.001** |
| MPV (fL) | 8.4 ± 0.9 | 8.4 ± 1.0 | 8.3 ± 0.4 | 0.645 |
| PDW% | 15.4 (11.5–15.9) | 15.4 (11.8–15.9) | 15.8 (6.3–16.1) | 0.836 |
| PCT% | 0.13 (0.1–0.2) | 0.1 (0.1–0.2) | 0.1 (0.1–0.2) | 0.225 |
| Ferritin (ng/mL) | 362.3 (273.1–399.9) | 336.2 (249.9–386.5) | 473.1 (428.3–496.0) | **<0.001** |

N = Number of participants; COVID-19 = Coronavirus disease-2019; Hb = Haemoglobin, **Males = 13–17, Females = 12–16 g/dL;** RBC = Red Blood Cell, **4.50–6.50 ×$10^{12}$/L;** HCT = Haematocrit, **40.0–54.0%;** MCV = Mean Cell Volume, **80–100 fL;** MCH = Mean Corpuscular Haemoglobin, **27–32 pg;** MCHC = Mean Corpuscular Haemoglobin Concentration, **32.0–36.0 g/dL;** RDW-CV = Red Cell Distribution Width-Coefficient of Variation, **11.0–16.0%;** TWBC = Total White Blood Cell, **4.0– 11.0 ×$10^9$/L;** Lymph# = Absolute Lymphocyte count, **1.00–5.00 ×$10^9$/L;** Gran# = Absolute Granulocyte count, **2.00–9.00 ×$10^9$/L;** MPV = Mean Platelet Volume, **6.0– 12.0 fL;** PLT = Platelets, **150–450 ×$10^9$/L;** PDW = Platelet Distribution Width, **11.0–18.0%;** PCT = Plateletcrit, **0.150–0.500%; ferritin, Males = 12–300, Females = 10– 150 ng/mL;** g/dL = Grams per decilitre; fL = Femtolitre; pg = Picogram. Parametric data were presented in mean ± standard deviation and compared with Student's test-test, whilst non-parametric data presented in medians (25th-75th percentile) were compared with Mann-Whitney U-test. p<0.05 was considered statistically significant.

values at admission (before treatment). Also, serum ferritin levels (ng/mL): 242.2 (197.1– 302.1) vs 362.3 (273.1–399.9), *p*<0.001 of the COVID-19 participants were significantly lower at recovery compared to the levels on admission (before treatment vs at recovery) (Table 3).

## Changes in plasma plasminogen activator inhibitor-1 antigen levels before treatment and at recovery

Plasma PAI-1 Ag levels were assessed on admission, and at recovery. There were significant changes in the plasma PAI-1 Ag levels on admission (before treatment commenced) and at recovery. The median PAI-1 Ag levels were reduced after successful recovery [89.6 ng/mL (74.9–100.8)] compared to the value at admission [103.1 ng/mL (93.2–128.7)], and this was statistically significant (*p<0.001*) as shown in Fig 4.

## Changes in PAI-1 and haematological parameters before treatment and at recovery stratified by COVID-19 severity

Blood cell parameters of both severe and non-severe COVID-19 patients significantly improved after recovery from the disease. Similarly, both PAI-1 and ferritin relatively reduced after recovery regardless of the COVID-19 disease severity (Table 4).

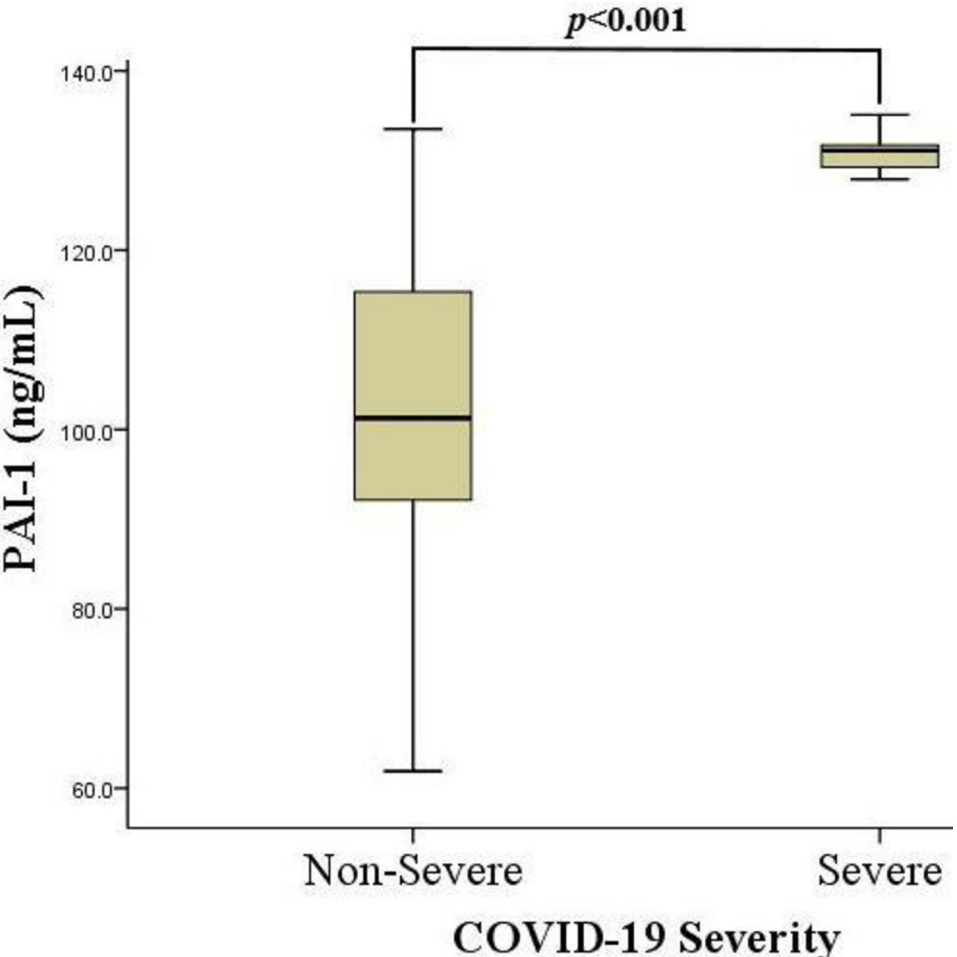

**Fig 3. Plasma plasminogen activator inhibitor-1 antigen levels and COVID-19 severity among the study participants.** PAI-1 = Plasminogen activator inhibitor-1, 2–46 ng/mL; ng/mL = Nanogram per millilitre. PAI-1 levels were compared using the Mann-Whitney U test. p< 0.05 was considered statistically significant.

## Discussion

COVID-19 patients present with life-threatening medical complications including coagulopathy and its related thrombotic events and deaths. Hypofibrinolysis, resulting from the upregulation of PAI-1 may be implicated in the development of thrombotic events in severe SARS-CoV-2-infected persons [7, 10]. This study determined changes in plasma PAI-1 Ag levels before treatment and during recovery from COVID-19.

Most of the COVID-19 participants in this study were females, and this is consistent with a previous finding in the United States (U.S.) [28]. This may be attributed to the increased exposure of females to the highly contagious SARS-CoV-2 as the transmission is often by air droplet, and females relatively interact more compared to males. However, this finding contradicts earlier studies in China [29] and India [30] where the majority of the COVID-19-infected participants were males. They attributed their findings to the higher amounts of Angiotensin-Converting Enzyme-2 (ACE-2) and Transmembrane Serine Protease-2 (TMPRSS-2) in males as well as hormonal influences on immune response [31]. SARS-CoV-2 cell entry is facilitated by spike protein S1 binding to the ACE-2 receptor, and protein priming by the host's TMPRSS2. This process requires S protein cleavage at the S1/S2 and the S2' site for the S2

**Table 3. Changes in full blood counts and serum ferritin levels before treatment and at recovery.**

| Variables | COVID-19 Participants | | P-value |
|---|---|---|---|
| | Before Treatment | At Recovery | |
| Hb (g/dL) | 11.4 (8.8–12.3) | 12.4 (11.5–13.6) | **<0.001** |
| RBC x $10^{12}$/L | 3.3 (2.9–4.0) | 4.3 (3.4–4.6) | **0.001** |
| HCT% | 33.0 ± 7.4 | 35.7 ± 6.2 | **0.048** |
| MCV (fL) | 97.5 ± 11.4 | 89.9 ± 12.1 | **0.025** |
| MCH (pg) | 32.5 (28.3–37.1) | 30.2 (26.8–34.7) | **0.016** |
| MCHC (g/dL) | 34.5 (33.7–35.8) | 34.3 (33.3–35.7) | 0.476 |
| RDW-CV% | 14.6 (13.6–15.6) | 14.2 (13.7–15.4) | 0.379 |
| TWBC x $10^9$/L | 5.7 (4.2–9.7) | 6.7 (5.9–8.0) | 0.862 |
| Lymph# x $10^9$/L | 3.3 (2.1–6.2) | 1.9 (1.3–2.1) | **<0.001** |
| Gran# x $10^9$/L | 2.3 ± 1.0) | 4.6 ± 1.8 | **<0.001** |
| MID# x $10^9$/L | 0.4 (0.3–0.6) | 0.5 (0.4–0.7) | 0.400 |
| PLT x $10^9$/L | 135.0 (107.0–193.0) | 229.0 (166.0–270.0) | **<0.001** |
| MPV (fL) | 8.4 ± 0.9 | 8.2 ± 0.9 | 0.190 |
| PDW% | 15.4 (11.5–15.9) | 15.6 (15.4–15.8) | **0.014** |
| PCT% | 0.13 (0.1–0.2) | 0.2 (0.1–0.2) | **0.030** |
| Ferritin (ng/mL) | 362.3 (273.1–399.9) | 242.2 (197.1–302.1) | **<0.001** |

COVID-19 = Coronavirus disease-2019; Hb = Haemoglobin, **Males = 13–17, Females = 12–16 g/dL;** RBC = Red Blood Cell, **4.50–6.50 ×$10^{12}$/L;** HCT = Haematocrit, **40.0–54.0%;** MCV = Mean Cell Volume, **80–100 fL;** MCH = Mean Corpuscular Haemoglobin, **27–32 pg;** MCHC = Mean Corpuscular Haemoglobin Concentration, **32.0–36.0 g/dL;** RDW-CV = Red Cell Distribution Width-Coefficient of Variation, **11.0–16.0%;** TWBC = Total White Blood Cell, **4.0–11.0 ×$10^9$/L;** Lymph# = Absolute Lymphocyte count, **1.00–5.00 ×$10^9$/L;** Gran# = Absolute Granulocyte count, **2.00–9.00 ×$10^9$/L;** MPV = Mean Platelet Volume, **6.0–12.0 fL;** PLT = Platelets, **150–450 ×$10^9$/L;** PDW = Platelet Distribution Width, **11.0–18.0%;** PCT = Plateletcrit, **0.150–0.500%; ferritin, Males = 12–300, Females = 10–150 ng/mL;** g/dL = Grams per decilitre; fL = Femtolitre; pg = Picogram. Parametric data were presented in mean ± standard deviation and compared with Paired Sample T-test, whilst non-parametric data presented in medians (25th-75th percentile) were compared with Wilcoxon signed-rank test. p<0.05 was considered statistically significant.

subunits to fuse the viral envelope [30]. The major age group (30–39 years) identified in this study is consistent with the study by Al-Muhanna et al. [32] in Najaf Province, Iraq. This age group constitutes the most active youth and working age, increasing their exposure to SARS-CoV-2 viral aerosols through human contact.

The significantly higher body temperature, pulse rate and SBP observed among the severe COVID-19 participants in this study is similar to earlier findings [33, 34]. The elevation in the vital signs of the COVID-19 participants could be related to the enhanced inflammatory response during the infection, as this eventually increases vagal tone and decreases heart rate variability [33–35]. The surge in BP could result from the SARS-CoV-2 modulating the renin-angiotensin-aldosterone system [35]. Also, the reduced SpO2% detected in the severe COVID-19 participants could be due to the associated hypoxemic respiratory failure and severe lung destruction [36].

This study found a significant reduction in erythrocyte parameters (RBC, HB and HCT) in the severely ill COVID-19 participants and this is similar to findings from earlier studies [37–40]. COVID-19 induces an inflammatory response, resulting in the release of cytokines (cytokine storm) such as tumour necrosis factor-alpha (TNF-α), and interleukins (IL-1, and IL-6) [3, 41, 42]. The enhanced inflammation was vindicated by the detection of significantly higher

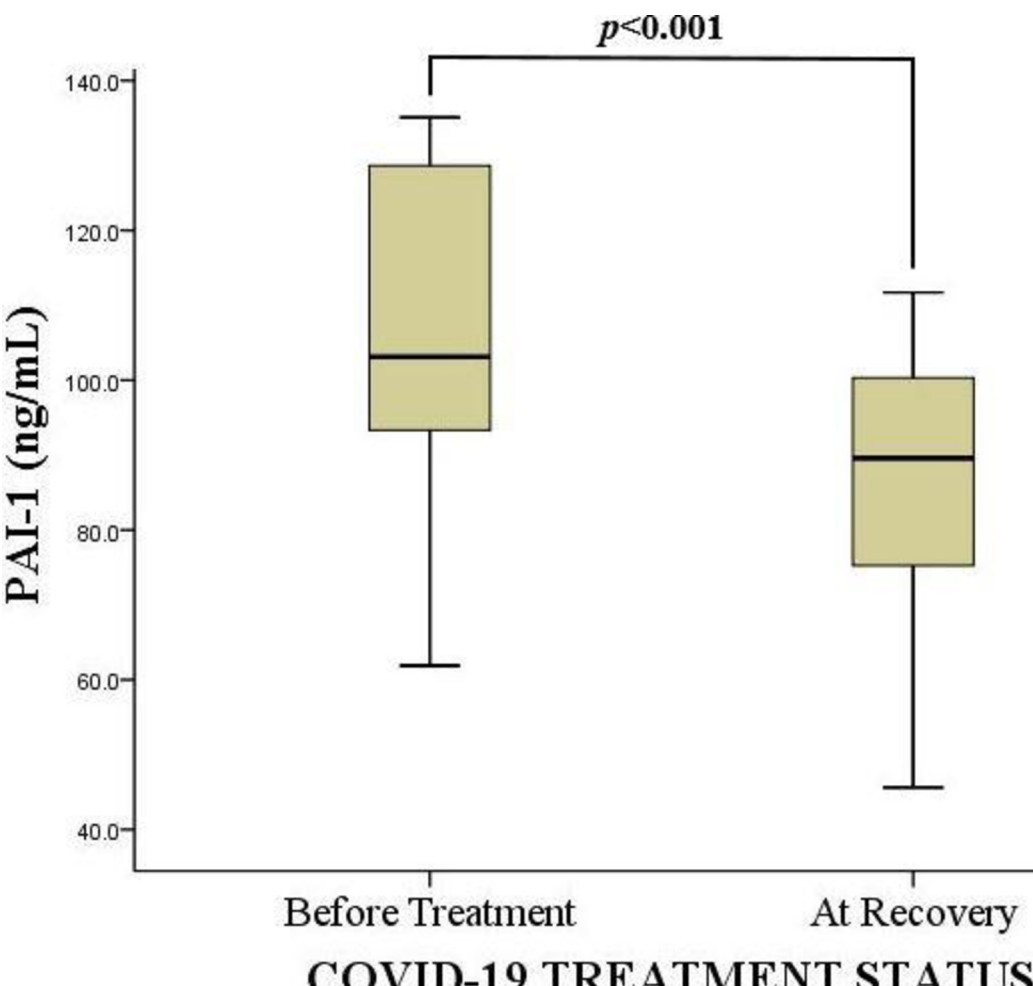

**Fig 4. Changes in plasma plasminogen activator inhibitor-1 antigen levels before treatment and at recovery.** PAI-1 Ag = Plasminogen activator inhibitor-1 antigen, **2–46 ng/mL**; ng/mL = Nano gram per deciliter. Wilcoxon signed-rank test was used to determine the significant difference in PAI-1 Ag levels before treatment and after COVID-19 recovery. $p < 0.05$ was considered statistically significant.

levels of serum ferritin among the severe COVID-19 participants in this study. Previous studies in Ghana [43], China [44], Iraq [45] and Mexico [19] also found increased ferritin levels in severe COVID-19 disease. Ferritin is an acute-phase reactant protein and increases during an inflammatory response, especially in the presence of anaemia. One reason for the reduction in erythrocyte parameters in severely ill COVID-19 participants is as a result of reduced erythropoiesis, contributed to by the inflammation [42]. The negative influence of inflammation on erythropoiesis occurs via a complex mechanism. Pro-inflammatory cytokines such as TNF-α and IL-1 suppress the erythropoietin (EPO) gene expression, resulting in a decreased production of the EPO and disturbed erythropoiesis [40, 42, 46]. Also, the invasion of SARS-CoV-2 triggers the innate immune system and induces the release of the inflammatory cytokine, IL-6 which triggers hepcidin release. Hepcidin, a hormone synthesized in hepatocytes, interferes with the intracellular iron transporter, ferroportin to regulate iron metabolism [3]. SARS-CoV-2-associated uncontrolled release of hepcidin and the hormone's interaction with ferroportin restrict iron availability for erythropoiesis leading to the development of functional iron deficiency anaemia. The insufficient delivery of oxygen ($O_2$) to tissues may contribute to the

**Table 4. Changes in PAI-1 and haematological parameters before treatment and at recovery stratified by COVID-19 severity.**

| Variables | Non-severe COVID-19 Patients (N = 40) | | | Severe COVID-19 Patients (N = 11) | | |
|---|---|---|---|---|---|---|
| | Before Treatment | After Recovery | *P-value* | Before Treatment | After Recovery | *P-value* |
| Hb (g/dL) | 11.8 (11.0–12.5) | 12.6 (11.5–13.6) | <**0.001** | 8.1 (7.3–8.4) | 11.7 (11.5–13.9) | **0.004** |
| RBC x 10$^{12}$/L | 3.4 (3.1–4.3) | 4.2 (3.5–4.6) | **0.034** | 2.9 (2.6–3.1) | 4.4 (3.2–4.8) | **0.004** |
| HCT% | 35.3 ± 6.7 | 35.5 ± 5.7 | 0.850 | 24.8 ± 2.6 | 36.6 ±8.0 | **0.001** |
| MCV (fL) | 97.5 ± 11.4 | 89.4 ± 12.4 | **0.005** | 88.9 ± 11.4 | 91.6 ± 10.7 | 0.627 |
| MCH (pg) | 34.2 (29.0–37.7) | 29.4 (26.8–34.5) | **0.002** | 32.4 (27.1–34.4) | 30.5 (26.9–36.1) | 0.476 |
| MCHC (g/dL) | 34.6 (33.6–35.8) | 34.5 (33.1–35.9) | 0.420 | 34.4 (34.1–35.5) | 34.3 (33.8–35.3) | 0.894 |
| RDW-CV% | 14.3 (13.3–15.2) | 14.2 (13.9–15.5) | 0.073 | 15.4 (14.6–19.9) | 14.5 (13.4–15.4) | 0.266 |
| TWBC x 10$^9$/L | 5.4 (3.7–6.6) | 6.8 (5.9–8.3) | **0.007** | 11.6 (9.9–14.2) | 6.7 (4.9–7.3) | **0.003** |
| Lymph# x 10$^9$/L | 2.8 (1.9–3.7) | 1.9 (1.3–2.1) | <**0.001** | 7.6 (6.5–10.3) | 1.9 (1.5–2.1) | **0.003** |
| Gran# x 10$^9$/L | 2.0 ± 0.8 | 4.8 ± 1.9 | <**0.001** | 3.5 ± 0.8 | 4.0 ± 1.4 | 0.223 |
| MID# x 10$^9$/L | 0.4 (0.3–0.6) | 0.6 (0.4–0.7) | 0.123 | 0.6 (0.3–0.8) | 0.5 (0.4–0.7) | 0.212 |
| PLT x 10$^9$/L | 165.5 (115.1–210.3) | 230.5 (170.0–269.5) | **0.002** | 86.4 (62.2–91.8) | 198.0 (152.0–318.0) | **0.003** |
| MPV (fL) | 8.4 ± 1.0 | 8.2 ± 0.9 | 0.357 | 8.3 ± 0.4 | 7.9 ± 1.0 | 0.279 |
| PDW% | 15.4 (11.8–15.9) | 15.6 (15.5–18.9) | **0.012** | 15.8 (6.3–16.1) | 15.6 (15.4–15.9) | 0.477 |
| PCT% | 0.1 (0.1–0.2) | 0.2 (0.1–0.2) | 0.078 | 0.1 (0.1–0.2) | 0.2 (0.1–0.2) | 0.091 |
| Ferritin (ng/mL) | 336.2 (249.9–386.5) | 214.7 (167.3–285.8) | <**0.001** | 473.1 (428.3–496.0) | 321.2 (251.5–357.5) | **0.003** |
| PAI-1 (ng/mL) | 101.3 (92.0–116.8) | 88.2 (73.9–97.0) | <**0.001** | 131.1 (128.7–131.9) | 102.3 (96.3–109.4) | **0.003** |

COVID-19 = Coronavirus disease-2019; Hb = Haemoglobin, **Males = 13–17, Females = 12–16 g/dL;** RBC = Red Blood Cell, **4.50–6.50 ×10$^{12}$/L;** HCT = Haematocrit, **40.0–54.0%;** MCV = Mean Cell Volume, **80–100 fL;** MCH = Mean Corpuscular Haemoglobin, **27–32 pg;** MCHC = Mean Corpuscular Haemoglobin Concentration, **32.0–36.0 g/dL;** RDW-CV = Red Cell Distribution Width-Coefficient of Variation, **11.0–16.0%;** TWBC = Total White Blood Cell, **4.0–11.0 ×10$^9$/L;** Lymph# = Absolute Lymphocyte count, **1.00–5.00 ×10$^9$/L;** Gran# = Absolute Granulocyte count, **2.00–9.00 ×10$^9$/L;** MPV = Mean Platelet Volume, **6.0–12.0 fL;** PLT = Platelets, **150–450 ×10$^9$/L;** PDW = Platelet Distribution Width, **11.0–18.0%;** PCT = Plateletcrit, **0.150–0.500%; ferritin, Males = 12–300, Females = 10–150 ng/mL;** g/dL = Grams per decilitre; fL = Femtolitre; pg = Picogram; PAI-1 = Plasminogen activator inhibitor-1, **2–46 ng/mL**. Parametric data were presented in mean ± standard deviation and compared with Paired Sample T-test, whilst non-parametric data presented in medians (25$^{th}$-75$^{th}$ percentile) were compared with Wilcoxon signed-rank test. p<0.05 was considered statistically significant.

occurrence of multiple-organ damage during severe SARS-CoV-2 infection [40, 46]. Again, membrane protein oxidation and fragmentation as well as increased glycolytic intermediate levels such as nitric oxide (NO), in COVID-19 patients' RBCs limit their ability to transport and deliver oxygen to the tissues. When moving from the lungs through the bloodstream, RBCs from COVID-19 patients may not be able to adapt to environmental changes in haemoglobin oxygen saturation, which could impair their ability to transport and deliver oxygen [47]. Mortaz et al. [48] found that the levels of NO in RBC were higher in COVID-19 patients than in non-COVID-19 hypoxemic patients, which might result in life-threatening vasoplegia and hypotension. However, the mechanism(s) underlying this accumulation of intracellular NO in COVID-19 patients' RBC remains unclear.

The relatively high total WBCs and lymphocytes among the participants with severe COVID-19 compared to their counterparts with the non-severe form of the disease is similar to previous findings [49, 50]. The increased leukocytes and lymphocytes in the severe COVID-19 participants may be due to the virus' ability to stimulate the immune response to recruit abundant immune cells. This finding however, contradicts earlier studies elsewhere which particularly identified lymphopaenia among severe SARS-CoV-2 infected persons [39, 43, 49, 51]. The differences in the findings may be due to the differences in the selection of the study subjects. Whiles most of the study participants in the current study had non-severe COVID-19,

the previous studies mostly recruited patients who were critically ill and had been admitted to the intensive care unit (ICU).

Thrombocytopaenia has been observed in severe COVID-19 patients in Ethiopia [52–54]; we observed similar findings in this study. SARS-CoV-2 may suppress bone marrow haematopoiesis by binding to particular receptors, reducing the primary production of platelets and causing thrombocytopaenia [55–57]. Similar findings were seen approximately a decade ago in participants who were infected with severe acute respiratory syndrome and the Middle East respiratory syndrome [58]. The virus' interaction with megakaryocytes and consequent decrease in platelet synthesis may contribute to the cause of the related thrombocytopaenia in COVID patients [59, 60]. Infection with COVID-19 may affect lung tissues and endothelial cells, which can lead to platelet aggregations, the development of microthrombi, and further platelet consumption [61, 62]. Additionally, SARS-CoV-2 infection may stimulate autoantibodies and immunological complexes, and enhance the immune destruction of thrombocytes [63].

The relatively higher plasma PAI-1 Ag level [131.1 ng/mL (128.7–131.9)] identified in the severe COVID-19 participants corresponds with earlier findings from the study by Nougier et al. where PAI-1 levels among COVID-19 patients in ICU was 96.3 ± 35 ng/mL, compared to the non-severe group, 76.8 ± 40 ng/mL [14]. Again, another study reported a higher plasma PAI-1 Ag levels when they compared to apparently healthy individuals without COVID-19 in their study [103.1 ng/mL (93.2–128.7)] of the COVID-19 patients observed in the present study is relatively higher than [64]. The earlier studies associated the disturbed fibrinolytic activities among moderately and critically ill SARS-CoV-2-infected patients with high PAI-1 levels [14, 64]. This may be attributed to the extent of the associated inflammatory response (cytokine storm) during COVID-19 [19, 44]. The continuous excessive cytokines release triggers inflammatory injury to the endothelium, resulting in increased expression of PAI-1, or may induce the release of the anti-fibrinolytic agent from activated platelets [7]. High levels of PAI-1 expression in other cell types such as macrophages have also been documented earlier [65]. The increased PAI-1 expression among severely ill COVID-19 subjects contributes to the hypercoagulability of the condition, as PAI-1 independently inhibits tissue plasminogen activator and contributes significantly to the development of thrombotic events [10].

Significant changes occurred following successful management and recovery from both severe and non-severe COVID-19. Red blood cells were elevated at recovery compared to the values before treatment in this study, and this is similar to recent studies [37, 40, 46]. This could be due to the decline in inflammatory response as vindicated by the reduced ferritin levels after recovery and the effective renal secretion of EPO hormone for erythropoiesis [40, 42, 46, 66]. The enhanced activation, aggregation, and trapping of the platelet caused by SARS-CoV-2-associated lung tissue damage eventually subsided following a reduction in the viral load and suppression of immune response. This eventually could reduce the consumptive coagulopathy seen in the severe form of the disease, and restore the normal platelets numbers.

The present study found significantly higher plasma PAI-1 Ag level among the COVID-19 treatment-naïve participants [103.1 ng/mL (93.2–128.7)] compared to the values after recovery from the disease [89.6 ng/mL (74.9–100.8)]. A cohort study by von Meijenfeldt et al. at Danderyd Hospital Stockholm, Sweden, that recruited 52 COVID-19 patients found a very low PAI-1 levels [3.15 (0.85–5.58)] after four months of follow-up [67]. The decrease in plasma PAI-1 Ag level after treatment may be a result of the restoration of the damaged endothelial tissue as well as the reduction in the hyperactivation of platelets.

This study could not assess the entire fibrinolytic system of the study participants. Also, the polymorphisms in the PAI-1 gene of the study participants could not be studied.

## Conclusion

Plasma PAI-1 Ag level was higher among severe COVID-19 participants. The COVID-19-associated inflammation could contribute to the reduction in red blood cell parameters and platelets. Successful recovery from COVID-19, with reduced inflammatory response as observed in the decline of serum ferritin levels restored the haematological parameters. Plasma levels of PAI-1 should be assessed during the management of severe COVID-19 in Ghana. Further studies to assess the entire fibrinolytic system of COVID-19 patients are recommended. This will enhance the early detection of probable thrombotic events and prompts Physicians to provide interventions to prevent thrombotic complications associated with COVID-19.

## Acknowledgments

Authors appreciate the enormous contributions of senior members of the Department of Biomedical Laboratory Sciences and the Department of Haematology, School of Allied Health Sciences, University for Development Studies, Tamale, Ghana. We appreciate the support of the staff of the Clinical Laboratory Departments of Sunyani Regional Hospital and Ankaase Methodist Hospital, Ghana, for their immense support. A big thank you to all participants in the study.

## Author Contributions

**Conceptualization:** Charles Nkansah, Michael Owusu, Samuel Kwasi Appiah, Kofi Mensah, Simon Bannison Bani, Lawrence Duah Agyemang, Ezekiel Bonwin Ackah, Yeduah Quansah, Caleb Paul Eshun, Abdul-Waliu Iddrisu, Abidatu Mohammed.

**Data curation:** Lawrence Duah Agyemang, Benjamin Twum, Wendy Akomeah Gyasi, Seth Anane, Reginald Akwasi Yeboah Antwi.

**Formal analysis:** Charles Nkansah, Felix Osei-Boakye.

**Investigation:** Lawrence Duah Agyemang, Yeduah Quansah, David Amoah Afrifa, Benjamin Twum.

**Methodology:** Charles Nkansah, Lawrence Duah Agyemang, Ezekiel Bonwin Ackah, Yeduah Quansah, David Amoah Afrifa, Caleb Paul Eshun, Abidatu Mohammed, Peace Esenam Agbadza, Candy Adwoa Ewusiwaa Wilson.

**Resources:** Ezekiel Bonwin Ackah, Yeduah Quansah, Caleb Paul Eshun, Abdul-Waliu Iddrisu, Abidatu Mohammed.

**Supervision:** Charles Nkansah, Michael Owusu.

**Validation:** Charles Nkansah, Lawrence Duah Agyemang, Ezekiel Bonwin Ackah, Gabriel Abbam, Samira Daud, Charles Angnataa Derigubah, Francis Atoroba Apodola, Selina Mintaah, Eugene Mensah Agyare, Wendy Akomeah Gyasi, Prince Antwi, Reginald Akwasi Yeboah Antwi.

**Visualization:** Charles Nkansah, Felix Osei-Boakye, Ezekiel Bonwin Ackah, Yeduah Quansah, Valentine Ayangba.

**Writing – original draft:** Charles Nkansah, Michael Owusu, Samuel Kwasi Appiah, Kofi Mensah, Simon Bannison Bani, Felix Osei-Boakye, Lawrence Duah Agyemang, Ezekiel Bonwin Ackah, Gabriel Abbam, Yeduah Quansah, Charles Angnataa Derigubah, Francis Atoroba Apodola, Valentine Ayangba, David Amoah Afrifa, Selina Mintaah, Eugene Mensah

Agyare, Wendy Akomeah Gyasi, Peace Esenam Agbadza, Candy Adwoa Ewusiwaa Wilson, Seth Anane, Prince Antwi.

**Writing – review & editing:** Charles Nkansah, Michael Owusu, Samuel Kwasi Appiah, Kofi Mensah, Simon Bannison Bani, Felix Osei-Boakye, Lawrence Duah Agyemang, Ezekiel Bonwin Ackah, Gabriel Abbam, Samira Daud, Yeduah Quansah, Charles Angnataa Derigubah, Francis Atoroba Apodola, Valentine Ayangba, David Amoah Afrifa, Caleb Paul Eshun, Abdul-Waliu Iddrisu, Selina Mintaah, Benjamin Twum, Abidatu Mohammed, Eugene Mensah Agyare, Wendy Akomeah Gyasi, Peace Esenam Agbadza, Candy Adwoa Ewusiwaa Wilson, Seth Anane, Prince Antwi, Reginald Akwasi Yeboah Antwi.

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
