## [Decision Letter · Decision Letter 0]

8 Feb 2023

PGPH-D-22-01975

Changes in plasma plasminogen activator inhibitor-1 antigen during COVID-19 management: A prospective cross-sectional study in a Regional Hospital, Ghana

Dear Dr. Charles Nkanash,

Thank you for submitting your manuscript to PLOS Global Public Health. After careful consideration, we feel that it has merit but does not fully meet PLOS Global Public Health’s publication criteria as it currently stands. Therefore, we invite you to submit a revised version of the manuscript that addresses the points raised during the review process.

Please submit your revised manuscript by March 07, 2023. If you will need more time than this to complete your revisions, please reply to this message or contact the journal office at globalpubhealth@plos.org. Please include the following items when submitting your revised manuscript:

We look forward to receiving your revised manuscript.

Kind regards,

Srinivasa Rao Mutheneni, PhD

Academic Editor

Journal Requirements:

2. Please provide separate figure files in .tif or .eps format only and remove any figures embedded in your manuscript file. Please also ensure that all files are under our size limit of 10MB.

Additional Editor Comments (if provided):

Reviewers' comments:

Reviewer's Responses to Questions

**Comments to the Author**

1. Does this manuscript meet PLOS Global Public Health’s publication criteria? Is the manuscript technically sound, and do the data support the conclusions? The manuscript must describe methodologically and ethically rigorous research with conclusions that are appropriately drawn based on the data presented.

Reviewer #1: Yes

Reviewer #2: Yes

2. Has the statistical analysis been performed appropriately and rigorously?

Reviewer #1: Yes

Reviewer #2: Yes

3. Have the authors made all data underlying the findings in their manuscript fully available (please refer to the Data Availability Statement at the start of the manuscript PDF file)?

Reviewer #1: Yes

Reviewer #2: Yes

4. Is the manuscript presented in an intelligible fashion and written in standard English?

Reviewer #1: Yes

Reviewer #2: Yes

5. Review Comments to the Author

Reviewer #1: The manuscript is addressing an important aspect of the pathophysiological changes that occur in individuals infected with SARS_CoV2 virus and how they vary with the severity of the disease in which the information about it has been limited. However, much as my responses above have been OK but the following are the comments about the manuscript.

General comments:

1. Sentences are too long and some of them are mixed and therefore authors should ensure that they make short sentences without losing the meaning (see sentences 66-73)

2. Some sentences are not written in full and so authors need to check the grammar and the sentence construction especially in the results section

3. Authors are too many and what was the contribution of each of the authors in this work

Specific comments:

1. Title does not reflect the work done and so I suggest to change it to "Effects of COVID-19 disease on PAF-1 antigen, hematological parameters and vital signs during disease management: A prospective cross-sectional study in a regional Hospital in Ghana"

Line 57: insert antigen after word PAI-1

Line 66-74: Results should be presented as means (CI at 95% and then the p-values. Otherwise just presenting the p-values does not reflect good presentation of the results. Secondly at what confidence intervals were these p-values computed.

Line 76: Use past tense and mot present tense and so change “is” to “was”

Line 100: It is too long and it addresses the global burden of COVID-19 and then in Ghana. The sentence should be split up.

Line 103: Insert “is” after COVID-19

Line 104: Insert “and this virus” after (SARS-CoV2)

Line 108: Replace word “suggested” to “reported” and authors need to explain more on the “Cytokine storm”

Line 113: What are the main cytokines involved in the cytokine storm?

Line 117-121: Sentence is too long and need to be split up since the meaning is lost

Line 127: Sentence need to be paraphrased

Line 136: Which enzyme has not been studied? And suggest the authors replace “our” with “present”

Line 138: insert “of the COVID-19 disease in the infected individuals” after word “complications”

Line 140: Should be “Materials and methods” other than “Methodology”

Line 141: Should be “Study design and setting”

Line 142: Insert “that was conducted” after word “study”

Line 147: Replace “surrounded “with “surrounding”

Line 149: Can authors provide a map of the study site if possible where the study was conducted!!

Line 154: Authors should include a statement of the “Helsinki declaration on use of human subjects in research”

Line 152: Replace word “this” with “the”

Line 155: It should read “Study population”

Line 173: Authors insert a sub-section “Study participant selection criteria” and then also separate the “Inclusion and Exclusion criteria” What was the age ranges of the study participants that were supposed to be included in the study? What happened to those who refused to consent or those who were severely ill and could not sign the consent form? What about those who were below 18 years of age? What happed to those who died and were not able to get the second set of data or there was no death recorded among those who were recruited?

Line 177: Include “Data collection and sources of data”

Line 182: What are those commercial tubes and their names!!!

Line 183 -186: Sentence is too long and please split it up.

Line 187: What was the rationale of using 2 anticoagulants i.e. K2EDTA and sodium citrate??

Line 195: What do authors mean “on daily basis” when they had to samples to be collected? Please clarify.

Line 203 -204: WHO classifies the COVID-19 disease progression? Please use that classification

Line 212: Insert “FBC) after “counts”

Line 228: Which “Lysing agent was used”?

Line 243-251: Many tests were used. Of these tests, which test was performed on which data?

Line 255: the measure is percentage (%), I propose that authors when presenting their results be presented as “% (n) all through. Even some of the results presented are confusing line on line 255. I also suggest of normal reference values are included to see the variation form the normal during the disease progression and recovery. I also suggest that “units” be put in brackets for each parameter

Line 262-265: Please construct full sentences.

Tables of the results: Are these results presented as “means, CI, P-values? Or as single value measured for a single sample? See all the tables. Also see comment on Line 255

Line 279 – 294: Construct sentences for each parameter and not mixing them

Line 311: Figure 1 and figure 2 cannot be seen and if so those figures and not clearly labeled to be distinguished.

Discussion:

Sentences are too long and they need to be split up.

Line369: Sentence is not clear and it need to ne paraphrased

Conclusion:

Line 470: use past tense and not present tense

Line 479: Grammar typo and it should be corrected i.e. “appreciater”

Otherwise, it is a good manuscript, if the above comments are addressed.

Reviewer #2: The results of this important and robust cross sectional study contribute towards the growing body of evidence required for development of new therapeutic interventions for management of the COVID-19 pandemic. Research and development of new drugs to treat COVID-19 and related comorbid conditions is still work in progress, results from this study inform the current management policies and guidelines, and is critical for future large scale clinical research studies.

6. PLOS authors have the option to publish the peer review history of their article (what does this mean?). If published, this will include your full peer review and any attached files.

**Do you want your identity to be public for this peer review?** For information about this choice, including consent withdrawal, please see our Privacy Policy.

Reviewer #1: No

Reviewer #2: **Yes: **Gift Tafadzwa Chareka

---

## [Decision Letter · Decision Letter 1]

20 Mar 2023

PGPH-D-22-01975R1

Effects of COVID-19 disease on PAI-1 antigen, haematological parameters and vital signs during disease management: A prospective cross-sectional study in a regional Hospital in Ghana

Dear Dr. Charles Nkansah,

Thank you for submitting your manuscript to PLOS Global Public Health. After careful consideration, we feel that it has merit but does not fully meet PLOS Global Public Health’s publication criteria as it currently stands. Therefore, we invite you to submit a revised version of the manuscript that addresses the points raised during the review process.

EDITOR: Please insert comments here and delete this placeholder text when finished. Be sure to:

Indicate which changes you require for acceptance versus which changes you recommendAddress any conflicts between the reviews so that it's clear which advice the authors should followProvide specific feedback from your evaluation of the manuscript

Please ensure that your decision is justified on PLOS Global Public Health’s publication criteria and not, for example, on novelty or perceived impact.

We look forward to receiving your revised manuscript.

Kind regards,

Srinivasa Rao Mutheneni, PhD

Academic Editor

Journal Requirements:

Additional Editor Comments (if provided):

Reviewers' comments:

Reviewer's Responses to Questions

**Comments to the Author**

1. If the authors have adequately addressed your comments raised in a previous round of review and you feel that this manuscript is now acceptable for publication, you may indicate that here to bypass the “Comments to the Author” section, enter your conflict of interest statement in the “Confidential to Editor” section, and submit your "Accept" recommendation.

Reviewer #1: All comments have been addressed

Reviewer #3: All comments have been addressed

2. Does this manuscript meet PLOS Global Public Health’s publication criteria? Is the manuscript technically sound, and do the data support the conclusions? The manuscript must describe methodologically and ethically rigorous research with conclusions that are appropriately drawn based on the data presented.

Reviewer #1: Yes

Reviewer #3: Partly

3. Has the statistical analysis been performed appropriately and rigorously?

Reviewer #1: (No Response)

Reviewer #3: Yes

4. Have the authors made all data underlying the findings in their manuscript fully available (please refer to the Data Availability Statement at the start of the manuscript PDF file)?

Reviewer #1: Yes

Reviewer #3: Yes

5. Is the manuscript presented in an intelligible fashion and written in standard English?

Reviewer #1: Yes

Reviewer #3: Yes

6. Review Comments to the Author

Reviewer #1: Few comments observed as below:

1. Line 200: K2EDTA tubes for full blood counts (FBC) – tubes should be purple top vacutainers!!!!!

2. Line 204- 205: Blood samples in the plain tube (after sufficient clotting) and citrate tube were spun to separate the respective serum and plasma from the cells --- Centrifuged and Model of centrifuge and at what gram-force?

3. Table 1: For age and gender, the findings should be reported as % (N) and not vice versa as it is reported in the text.

Reviewer #3: (No Response)

7. PLOS authors have the option to publish the peer review history of their article (what does this mean?). If published, this will include your full peer review and any attached files.

**Do you want your identity to be public for this peer review?** For information about this choice, including consent withdrawal, please see our Privacy Policy.

Reviewer #1: No

Reviewer #3: **Yes: **James Nyabuga Nyariki

---

## [Decision Letter · Decision Letter 2]

10 Apr 2023

PGPH-D-22-01975R2

Effects of COVID-19 disease on PAI-1 antigen, haematological parameters and vital signs during disease management: A prospective cross-sectional study in a regional Hospital in Ghana

Dear Dr. Charles Nkansah,

Thank you for submitting your manuscript to PLOS Global Public Health. After careful consideration, we feel that it has merit but does not fully meet PLOS Global Public Health’s publication criteria as it currently stands. Therefore, we invite you to submit a revised version of the manuscript that addresses the points raised during the review process.

We look forward to receiving your revised manuscript.

Kind regards,

Srinivasa Rao Mutheneni, PhD

Academic Editor

Journal Requirements:

Additional Editor Comments (if provided):

Reviewers' comments:

Reviewer's Responses to Questions

**Comments to the Author**

1. If the authors have adequately addressed your comments raised in a previous round of review and you feel that this manuscript is now acceptable for publication, you may indicate that here to bypass the “Comments to the Author” section, enter your conflict of interest statement in the “Confidential to Editor” section, and submit your "Accept" recommendation.

Reviewer #1: All comments have been addressed

Reviewer #3: (No Response)

2. Does this manuscript meet PLOS Global Public Health’s publication criteria? Is the manuscript technically sound, and do the data support the conclusions? The manuscript must describe methodologically and ethically rigorous research with conclusions that are appropriately drawn based on the data presented.

Reviewer #1: Yes

Reviewer #3: Yes

3. Has the statistical analysis been performed appropriately and rigorously?

Reviewer #1: Yes

Reviewer #3: No

4. Have the authors made all data underlying the findings in their manuscript fully available (please refer to the Data Availability Statement at the start of the manuscript PDF file)?

Reviewer #1: Yes

Reviewer #3: Yes

5. Is the manuscript presented in an intelligible fashion and written in standard English?

Reviewer #1: Yes

Reviewer #3: Yes

6. Review Comments to the Author

Reviewer #1: Table 1: Information in column of Non-severe for female shifted and added itself to the male. Let it be rectified. Otherwise am okay with it now.

Reviewer #3: I have gone through the revised manuscript by Nkansah et al.., unfortunately none of the issues I had raised has been addressed. The current uploaded letter to the editor by the authors is titled “I am pleased to resubmit manuscript titled “Serum anti-erythropoietin antibodies among pregnant women with Plasmodium falciparum malaria and anaemia: A case-control study in northern Ghana “for your consideration”. “Whereas the current manuscript is talking about Effects of COVID-19 disease on PAI-1 antigen, haematological 1 parameters and vital signs during disease management: A prospective cross-sectional study in a regional Hospital in Ghana”.

In lieu to the forgoing it has become difficult from my side to give further recommendations on this manuscript.

7. PLOS authors have the option to publish the peer review history of their article (what does this mean?). If published, this will include your full peer review and any attached files.

**Do you want your identity to be public for this peer review?** For information about this choice, including consent withdrawal, please see our Privacy Policy.

Reviewer #1: **Yes: **Dr. Godfrey S. Bbosa

Reviewer #3: **Yes: **Dr. James nyabuga Nyariki

---

## [Decision Letter · Decision Letter 3]

27 Apr 2023

PGPH-D-22-01975R3

Effects of COVID-19 disease on PAI-1 antigen and haematological parameters during disease management: A prospective cross-sectional study in a regional Hospital in Ghana

Dear Dr. Charles Nkansah,

Thank you for submitting your manuscript to PLOS Global Public Health. After careful consideration, we feel that it has merit but does not fully meet PLOS Global Public Health’s publication criteria as it currently stands. Therefore, we invite you to submit a revised version of the manuscript that addresses the points raised during the review process.

We look forward to receiving your revised manuscript.

Kind regards,

Srinivasa Rao Mutheneni, PhD

Academic Editor

Journal Requirements:

Additional Editor Comments (if provided):

Reviewers' comments:

Reviewer's Responses to Questions

**Comments to the Author**

1. If the authors have adequately addressed your comments raised in a previous round of review and you feel that this manuscript is now acceptable for publication, you may indicate that here to bypass the “Comments to the Author” section, enter your conflict of interest statement in the “Confidential to Editor” section, and submit your "Accept" recommendation.

Reviewer #3: All comments have been addressed

Reviewer #4: (No Response)

2. Does this manuscript meet PLOS Global Public Health’s publication criteria? Is the manuscript technically sound, and do the data support the conclusions? The manuscript must describe methodologically and ethically rigorous research with conclusions that are appropriately drawn based on the data presented.

Reviewer #3: (No Response)

Reviewer #4: Yes

3. Has the statistical analysis been performed appropriately and rigorously?

Reviewer #3: Yes

Reviewer #4: Yes

4. Have the authors made all data underlying the findings in their manuscript fully available (please refer to the Data Availability Statement at the start of the manuscript PDF file)?

Reviewer #3: Yes

Reviewer #4: Yes

5. Is the manuscript presented in an intelligible fashion and written in standard English?

Reviewer #3: Yes

Reviewer #4: Yes

6. Review Comments to the Author

Reviewer #3: The statement statistical software for the data analysis (SPSS) and the significance level should be completely deleted from the abstract

Reviewer #4: (No Response)

7. PLOS authors have the option to publish the peer review history of their article (what does this mean?). If published, this will include your full peer review and any attached files.

**Do you want your identity to be public for this peer review?** For information about this choice, including consent withdrawal, please see our Privacy Policy.

Reviewer #3: **Yes: **JAMES NYABUGA NYARIKI

Reviewer #4: **Yes: **Sukanta Chowdhury

---

## [Decision Letter · Decision Letter 4]

31 May 2023

Effects of COVID-19 disease on PAI-1 antigen and haematological parameters during disease management: A prospective cross-sectional study in a regional Hospital in Ghana

PGPH-D-22-01975R4

Dear Charles Nkansah,

We are pleased to inform you that your manuscript 'Effects of COVID-19 disease on PAI-1 antigen and haematological parameters during disease management: A prospective cross-sectional study in a regional Hospital in Ghana' has been provisionally accepted for publication in PLOS Global Public Health.

Best regards,

Srinivasa Rao Mutheneni, PhD

Academic Editor

Reviewer Comments (if any, and for reference):

Reviewer's Responses to Questions

**Comments to the Author**

1. If the authors have adequately addressed your comments raised in a previous round of review and you feel that this manuscript is now acceptable for publication, you may indicate that here to bypass the “Comments to the Author” section, enter your conflict of interest statement in the “Confidential to Editor” section, and submit your "Accept" recommendation.

Reviewer #4: All comments have been addressed

2. Does this manuscript meet PLOS Global Public Health’s publication criteria? Is the manuscript technically sound, and do the data support the conclusions? The manuscript must describe methodologically and ethically rigorous research with conclusions that are appropriately drawn based on the data presented.

Reviewer #4: Yes

3. Has the statistical analysis been performed appropriately and rigorously?

Reviewer #4: Yes

4. Have the authors made all data underlying the findings in their manuscript fully available (please refer to the Data Availability Statement at the start of the manuscript PDF file)?

Reviewer #4: Yes

5. Is the manuscript presented in an intelligible fashion and written in standard English?

Reviewer #4: Yes

6. Review Comments to the Author

Reviewer #4: Your responses should be linked with line and page numbers. So, please add line number(s) for each revision in the future.

7. PLOS authors have the option to publish the peer review history of their article (what does this mean?). If published, this will include your full peer review and any attached files.

**Do you want your identity to be public for this peer review?** For information about this choice, including consent withdrawal, please see our Privacy Policy.

Reviewer #4: **Yes: **Sukanta Chowdhury
